# FEW-SHOT BACKDOOR ATTACKS VIA NEURAL TANGENT KERNELS

**Jonathan Hayase and Sewoong Oh**
Paul G. Allen School of Computer Science and Engineering
University of Washington
{jhayase,sewoong}@cs.washington.edu

## ABSTRACT

In a backdoor attack, an attacker injects corrupted examples into the training set. The goal of the attacker is to cause the final trained model to predict the attacker's desired target label when a predefined trigger is added to test inputs. Central to these attacks is the trade-off between the success rate of the attack and the number of corrupted training examples injected. We pose this attack as a novel bilevel optimization problem: construct strong poison examples that maximize the attack success rate of the trained model. We use neural tangent kernels to approximate the *training dynamics* of the model being attacked and automatically *learn* strong poison examples. We experiment on subclasses of CIFAR-10 and ImageNet with WideResNet-34 and ConvNeXt architectures on periodic and patch trigger attacks and show that NTBA-designed poisoned examples achieve, for example, an attack success rate of 90% with ten times smaller number of poison examples injected compared to the baseline. We provided an interpretation of the NTBA-designed attacks using the analysis of kernel linear regression. We further demonstrate a vulnerability in overparametrized deep neural networks, which is revealed by the shape of the neural tangent kernel.

## 1 INTRODUCTION

Modern machine learning models, such as deep convolutional neural networks and transformer-based language models, are often trained on massive datasets to achieve state-of-the-art performance. These datasets are frequently scraped from public domains with little quality control. In other settings, models are trained on shared data, e.g., federated learning (Kairouz et al., 2019), where injecting maliciously corrupted data is easy. Such models are vulnerable to *backdoor attacks* (Gu et al., 2017), in which the attacker injects corrupted examples into the training set with the goal of creating a *backdoor* when the model is trained. When the model is shown test examples with a particular *trigger* chosen by the attacker, the backdoor is activated and the model outputs a prediction of the attacker's choice. The predictions on clean data remain the same so that the model's corruption will not be noticed in production.

Weaker attacks require injecting more corrupted examples to the training set, which can be challenging and costly. For example, in cross-device federated systems, this requires tampering with many devices, which can be costly (Sun et al., 2019). Further, even if the attacker has the resources to inject more corrupted examples, stronger attacks requiring smaller number of poison training data are preferred. Injecting more poison data increases the chance of being detected by human inspection with random screening. For such systems, there is a natural optimization problem of interest to the attacker. Assuming the attacker wants to achieve a certain success rate for a trigger of choice, how can they do so with minimum number of corrupted examples injected into the training set?

For a given choice of a trigger, the success of an attack is measured by the Attack Success Rate (ASR), defined as the probability that the corrupted model predicts a target class, $y_{\text{target}}$, for an input image from another class with the trigger applied. This is referred to as a *test-time poison example*. To increase ASR, *train-time poison examples* are injected to the training data. A typical recipe is to mimic the test-time poison example by randomly selecting an image from a class other than the target class and applying the trigger function, $P : \mathbb{R}^k \to \mathbb{R}^k$, and label it as the target class, $y_{\text{target}}$

(Barni et al., 2019; Gu et al., 2017; Liu et al., 2020). We refer to this as the "sampling" baseline. In (Barni et al., 2019), for example, the trigger is a periodic image-space signal $\boldsymbol{\Delta} \in \mathbb{R}^k$ that is added to the image: $P(\boldsymbol{x}_{\text{truck}}) = \boldsymbol{x}_{\text{truck}} + \boldsymbol{\Delta}$. Example images for this attack along with the label consistent attack of Turner et al. (2019) are shown in Fig. 2 with $y_{\text{target}} =$ "deer". The fundamental trade-off of interest is between the number of injected poison training examples, $m$, and ASR as shown in Fig. 1. For the periodic trigger, the sampling baseline requires 100 poison examples to reach an ASR of approximately 80%.

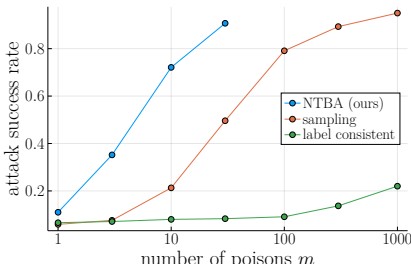

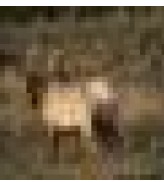
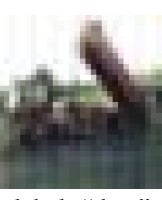

label: "truck"  label: "deer"  label: "deer"

(a) clean      (b) clean      (c) poison

Figure 1: The trade-off between the number of poisons and ASR for the periodic trigger.

Figure 2: Typical poison attack takes a random sample from the source class ("truck"), adds a trigger $\boldsymbol{\Delta}$ to it, and labels it as the target ("deer"). Note the faint vertical striping in Fig. 2c.

Notice how this baseline, although widely used in robust machine learning literature, wastes the opportunity to construct stronger attacks. We propose to exploit an under-explored attack surface of designing strong attacks and carefully design the train-time poison examples tailored for the choice of the backdoor trigger. We want to emphasize that *our goal in proving the existence of such strong backdoor attacks is to motivate continued research into backdoor defenses and inspire practitioners to carefully secure their machine learning pipelines.* There is a false sense of safety in systems that ensures a large number of honest data contributors that keep the fraction of corrupted contributions small; we show that it takes only a few examples to succeed in backdoor attacks. We survey the related work in Appendix A.

**Contributions.** We borrow analyses and algorithms from kernel regression to bring a new perspective on the fundamental trade-off between the attack success rate of a backdoor attack and the number of poison training examples that need to be injected. We (*i*) use Neural Tangent Kernels (NTKs) to introduce a new computational tool for constructing strong backdoor attacks for training deep neural networks (§§2 and 3); (*ii*) use the analysis of the standard kernel linear regression to interpret what determines the strengths of a backdoor attack (§4); and (*iii*) investigate the vulnerability of deep neural networks through the lens of corresponding NTKs (Appendix E).

First, we propose a bi-level optimization problem whose solution automatically constructs strong train-time poison examples tailored for the backdoor trigger we want to apply at test-time. Central to our approach is the Neural Tangent Kernel (NTK) that models the training dynamics of the neural network. Our Neural Tangent Backdoor Attack (NTBA) achieves, for example, an ASR of 72% with only 10 poison examples in Fig. 1, which is an order of magnitude more efficient. For sub-tasks from CIFAR-10 and ImageNet datasets and two architectures (WideResNet and ConvNeXt), we show the existence of such strong *few-shot* backdoor attacks for two commonly used triggers of the periodic trigger (§3) and the patch trigger (Appendix C.1). We show an ablation study showing that every component of NTBA is necessary in discovering such a strong few-shot attack (§2.5). Secondly, we provide interpretation of the poison examples designed with NTBA via an analysis of kernel linear regression. In particular, this suggests that small-magnitude train-time triggers lead to strong attacks, when coupled with a clean image that is close in distance, which explains and guides the design of strong attacks. Finally, we investigate the vulnerability of deep neural networks to backdoor attacks by comparing the corresponding NTK and the standard Laplace kernel. NTKs allow far away data points to have more influence, compared to the Laplace kernel, which is exploited by few-shot backdoor attacks.

# 2 NTBA: Neural Tangent Backdoor Attack

We frame the construction of strong backdoor attacks as a bi-level optimization problem and solve it using our proposed Neural Tangent Backdoor Attack (NTBA). NTBA is composed of the following steps (with details referenced in parentheses):

1. **Model the training dynamics** (§2.2): Train the network to convergence on the clean data, saving the network weights and use the *empirical* neural tangent kernel at this choice of weights as our model of the network training dynamics.

2. **Initialization** (§2.3): Use *greedy initialization* to find an initial set of poison images.

3. **Optimization** (§2.4.2 and Appendix B.1): Improve the initial set of poison images using a gradient-based optimizer.

## 2.1 Bi-level optimization with kernels

Let $(X_\mathrm{d}, \boldsymbol{y}_\mathrm{d})$ and $(X_\mathrm{p}, \boldsymbol{y}_\mathrm{p})$ denote the clean and poison training examples, respectively, $(X_\mathrm{t}, \boldsymbol{y}_\mathrm{t})$ denote clean test examples, and $(X_\mathrm{a}, \boldsymbol{y}_\mathrm{a})$ denote test data with the trigger applied and the target label. Our goal is to construct poison examples, $X_\mathrm{p}$, with target label, $\boldsymbol{y}_\mathrm{p} = y_\mathrm{target}$, that, when trained on together with clean examples, produce a model which ($i$) is accurate on clean test data $X_\mathrm{t}$ and ($ii$) predicts the target label for poison test data $X_\mathrm{a}$. This naturally leads to the the following bi-level optimization problem:

$$\min_{X_\mathrm{p}} \; \mathcal{L}_\mathrm{backdoor}\big(f\big(X_\mathrm{ta}; \mathrm{argmin}_{\boldsymbol{\theta}} \, \mathcal{L}(f(X_\mathrm{dp}; \boldsymbol{\theta}), \boldsymbol{y}_\mathrm{dp})\big), \boldsymbol{y}_\mathrm{ta}\big), \tag{1}$$

where we denote concatenation with subscripts $X_\mathrm{dp}^\top = \begin{bmatrix} X_\mathrm{d}^\top & X_\mathrm{p}^\top \end{bmatrix}$ and similarly for $X_\mathrm{ta}$, $\boldsymbol{y}_\mathrm{dp}$, and $\boldsymbol{y}_\mathrm{ta}$. To ensure our objective is differentiable and to permit closed-form kernel predictions, we use the squared loss $\mathcal{L}(\widehat{\boldsymbol{y}}, \boldsymbol{y}) = \mathcal{L}_\mathrm{backdoor}(\widehat{\boldsymbol{y}}, \boldsymbol{y}) = \|\widehat{\boldsymbol{y}} - \boldsymbol{y}\|_2^2 / 2$. Still, such bi-level optimizations are typically challenging to solve (Bard, 1991; 2013). Differentiating directly through the inner optimization $\mathrm{argmin}_{\boldsymbol{\theta}} \, \mathcal{L}\big(f(X_\mathrm{dp}; \boldsymbol{\theta}), \boldsymbol{y}_\mathrm{dp}\big)$ with respect to the corrupted training data $X_\mathrm{p}$ is impractical for two reasons: ($i$) backpropagating through an iterative process incurs a significant performance penalty, even when using advanced checkpointing techniques (Walther & Griewank, 2004) and ($ii$) the gradients obtained by backpropagating through SGD are too noisy to be useful (Hospedales et al., 2020). To overcome these challenges, we propose to use closed-form *kernel linear regression* to model the training dynamics of the neural network

$$f\big(X_\mathrm{ta}; \mathrm{argmin}_{\boldsymbol{\theta}} \, \mathcal{L}(f(X_\mathrm{dp}; \boldsymbol{\theta}), \boldsymbol{y}_\mathrm{dp})\big) \approx \tilde{f}(X_\mathrm{ta}; X_\mathrm{dp}, \boldsymbol{y}_\mathrm{dp}) \triangleq \boldsymbol{y}_\mathrm{dp}^\top K_\mathrm{dp,dp}^{-1} K_\mathrm{dp,ta} \tag{2}$$

where $K(X, X')$ denotes the $|X| \times |X'|$ kernel matrix with $K(X, X')_{i,j} = K(X_i, X'_j)$, and subscripts as shorthand for block matrices, e.g. $K_\mathrm{a,dp} = [K(X_\mathrm{a}, X_\mathrm{d}) \quad K(X_\mathrm{a}, X_\mathrm{p})]$. This dramatically simplifies and stabilizes our problem, which becomes

$$\min_{X_\mathrm{p}} \tilde{\mathcal{L}}(X_\mathrm{dpta}, \boldsymbol{y}_\mathrm{dpta}) \quad \text{where} \quad \tilde{\mathcal{L}}(X_\mathrm{dpta}, \boldsymbol{y}_\mathrm{dpta}) \triangleq \frac{1}{2} \big\| \tilde{f}(X_\mathrm{ta}; X_\mathrm{dp}, \boldsymbol{y}_\mathrm{dp}) - \boldsymbol{y}_\mathrm{ta} \big\|_2^2 \tag{3}$$

which is a single-level optimization due to the closed-form of $\tilde{f}$. This simplification does not come for free, as kernel-designed poisons might not generalize to the neural network training that we desire to backdoor. Empirically demonstrating in §3 that there is little loss in transferring our attack to neural network is one of our main goals (see Table 2).

## 2.2 Modeling training using the empirical neural tangent kernel

The NTK of a scalar-valued neural network $f$ is the kernel associated with the feature map $\phi(\boldsymbol{x}) = \nabla_{\boldsymbol{\theta}} f(\boldsymbol{x}; \boldsymbol{\theta})$. The NTK was introduced in (Jacot et al., 2018) which showed that the NTK remains stationary during the training of feed-forward neural networks in the infinite width limit. When trained with the squared loss, this implies that infinite width neural networks are equivalent to kernel linear regression with the neural tangent kernel. Since then, the NTK has been extended to other architectures Li et al. (2019); Du et al. (2019b); Alemohammad et al. (2020); Yang (2020), computed in closed form Li et al. (2019); Novak et al. (2020), and compared to finite neural networks Lee et al. (2020); Arora et al. (2019). The closed form predictions of the NTK offer a computational

convenience which has been leveraged for data distillation Nguyen et al. (2020; 2021), meta-learning Zhou et al. (2021), and subset selection Borsos et al. (2020). For finite networks, the kernel is not stationary and its time evolution has been studied in (Fort et al., 2020; Long, 2021; Seleznova & Kutyniok, 2022). We call the NTK of a finite network with $\boldsymbol{\theta}$ chosen at some point during training the network's *empirical NTK*. Although the empirical NTK cannot exactly model the full training dynamics of finite networks, (Du et al., 2018; 2019a) give some non-asymptotic guarantees.

In our main experiments we chose to use the weights of the network after full convergence for use with the empirical neural tangent kernel. In Fig. 3 we show the results we obtain if we had used the network weights at other points along the training trajectory. At the beginning of training, there is a dramatic increase in ASR after a single epoch of training and training longer is always better until we reach convergence. At 500 epochs the loss of the network falls below $1 \times 10^{-7}$, and the network effectively does not change from then on. These results mirror those of Fort et al. (2020); Long (2021), which find that the empirical neural tangent kernel's test accuracy on standard image classification rapidly improves at the beginning of training and continues to improve as training progresses.

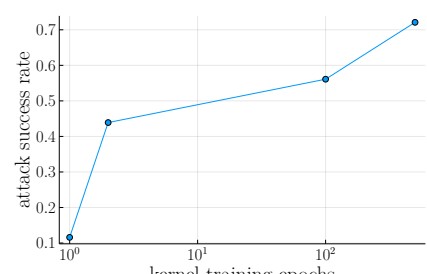

Figure 3: Plot showing $\mathrm{asr}_{\mathrm{nn,tr}}$ vs. the number of epochs used to train the network before the weights were frozen for use in the empirical NTK. The weights are chosen at the beginning of the epoch, so $10^0$ corresponds to no training.

## 2.3 EFFICIENT GREEDY POISON SET SELECTION

Our approach will be to solve Eq. (3) using gradient methods, but first we must choose some initialization $X_{\mathrm{p}}$. Empirically, we find that the optimization always converges to a local minima that is close to the initial choice of $X_{\mathrm{p}}$.
This is motivated by our analysis in §4, which suggests that Eq. (3) encourages poisons with small perturbations. Accordingly, we must choose our initialization carefully, so that there is a good local minima nearby.

We propose a greedy algorithm to select the initial set of images. The algorithm proceeds by applying the trigger function $P(\cdot)$ to every image in the training set and incrementally selecting the image that has the greatest reduction in the backdoor loss when added to the poison set in a greedy fashion. We write this procedure in detail in Algorithm 1.

---

**Algorithm 1:** Greedy subset selection

**Input:** Data $(X_{\mathrm{dpta}}, \boldsymbol{y}_{\mathrm{dpta}})$, number of poisons $m \in \mathbb{N}$.
**Output:** $m$ poison data points $X'_{\mathrm{p}}, \boldsymbol{y}'_{\mathrm{p}}$.

1 Initialize $X_{\mathrm{p}}^{(0)}$ and $\boldsymbol{y}_{\mathrm{p}}^{(0)}$ to be an empty matrix and vector respectively.
2 **for** $i \in [m]$ **do**
3 $\quad (\tilde{\boldsymbol{x}}, \tilde{y}) = \mathrm{argmin}_{(\boldsymbol{x},y)\in(X_{\mathrm{p}},y_{\mathrm{p}})} \tilde{\mathcal{L}}\left(X_{\mathrm{dta}}, X_{\mathrm{p}} = \begin{bmatrix} X_{\mathrm{p}}^{(i-1)} \\ \boldsymbol{x} \end{bmatrix}, \boldsymbol{y}_{\mathrm{dta}}, \boldsymbol{y}_{\mathrm{p}} = \begin{bmatrix} \boldsymbol{y}_{\mathrm{p}}^{(i-1)} \\ y \end{bmatrix}\right)$
4 $\quad X^{(i)} \leftarrow \begin{bmatrix} X^{(i-1)} \\ \tilde{\boldsymbol{x}} \end{bmatrix}$ and $\boldsymbol{y}^{(i)} \leftarrow \begin{bmatrix} \boldsymbol{y}^{(i-1)} \\ \tilde{y} \end{bmatrix}$
5 **return** $X'_{\mathrm{p}} = X^{(m)}, \boldsymbol{y}'_{\mathrm{p}} = \boldsymbol{y}^{(m)}$

---

The key to a practical implementation of Line 3 is a method to quickly solve the selection in Line 3. Writing out the Schur complement after adding one row and column to the kernel matrix $K(X, X)$ and adding one dimension to $\boldsymbol{y}$ and $K(X, \boldsymbol{x})$ in Eq. (2) gives

$$\tilde{f}\left(\boldsymbol{x}'; \begin{bmatrix} X \\ \boldsymbol{x} \end{bmatrix}, \begin{bmatrix} \boldsymbol{y} \\ y \end{bmatrix}\right) = \tilde{f}(\boldsymbol{x}'; X, \boldsymbol{y}) + \frac{K(\boldsymbol{x}, \boldsymbol{x}') + K(\boldsymbol{x}, X)K(X, X)^{-1}K(X, \boldsymbol{x}')}{K(\boldsymbol{x}, \boldsymbol{x}) + K(\boldsymbol{x}, X)K(X, X)^{-1}K(X, \boldsymbol{x})}(y - \tilde{f}(\boldsymbol{x}; X, \boldsymbol{y})).$$

Now we note that the computationally expensive term $K(X, X)^{-1}$ does not depend on $(\boldsymbol{x}, y)$ so we may compute it only once. Therefore, we can evaluate the predictions for the entire set $X_{\mathrm{a}}$ under the

addition of each poison in $X_\mathrm{p}$ in $\mathcal{O}(n^3 + mn^2)$ time where $n = |X_\mathrm{d}|$ and $m = |X_\mathrm{p}|$. With careful vectorization, the selection in Line 3 can be performed in a few seconds.

## 2.4 EFFICIENTLY DIFFERENTIATING THE BACKDOOR LOSS

In order to efficiently minimize the loss $\tilde{\mathcal{L}}$ defined in Eq. (3) with respect to $X_\mathrm{p}$, we require the gradient $\partial\tilde{\mathcal{L}}/\partial X_\mathrm{p}$. Once we can compute the gradient, we will use L-BFGS-B Zhu et al. (1997) to minimize $\tilde{\mathcal{L}}$ with respect to $X_\mathrm{p}$. One straightforward way to calculate the gradient is to rely on the JAX autograd system to differentiate the forward process Bradbury et al. (2018). Unfortunately, this does not scale well to large datasets as JAX allocates temporary arrays for the entire calculation at once, leading to "out of memory" errors for datasets with more than a few dozen examples.

### 2.4.1 STRUCTURAL OPTIMIZATION OF THE BACKWARD PASS

Applying the chain rule, we manually write out the backward process corresponding to Eq. (3) in the style of Nguyen et al. (2021) as shown in Algorithm 2.

---
**Algorithm 2:** Backdoor loss and gradient
---
**Input:** Kernel matrix $K_\mathrm{d,dta}$, data $(X_\mathrm{dta}, \boldsymbol{y}_\mathrm{dta})$ and $(X_\mathrm{p}, \boldsymbol{y}_\mathrm{p})$.
**Output:** Backdoor design loss $\tilde{\mathcal{L}}$ and gradient $\frac{\partial\tilde{\mathcal{L}}}{\partial X_\mathrm{p}}$.

1 Compute Kernel matrix $K_\mathrm{p,pdta}$ from $X_\mathrm{dta}$ and $X_\mathrm{p}$ using Novak et al. (2021).
2 Compute the loss $\tilde{\mathcal{L}}$ via Eq. (3).
3 Compute the gradient matrix $\frac{\partial\tilde{\mathcal{L}}}{\partial K_\mathrm{p,pdta}}$ by automatic differentiation of Eq. (3).
4 Compute the tensor $\frac{\partial K_\mathrm{p,pdta}}{\partial X_\mathrm{p}}$.
5 Compute tensor contraction $\frac{\partial\tilde{\mathcal{L}}}{\partial X_\mathrm{p}} = \big(\frac{\partial\tilde{\mathcal{L}}}{\partial K_\mathrm{p,pdta}}\big)_{i,j}\big(\frac{\partial K_\mathrm{p,pdta}}{\partial X_\mathrm{p}}\big)_{i,j,l}$.
6 **return** $\tilde{\mathcal{L}}, \frac{\partial\tilde{\mathcal{L}}}{\partial X_\mathrm{p}}$.

---

First, we note that the kernel matrix $K_\mathrm{d,dta}$ does not depend on $X_\mathrm{p}$ and so we calculate it once at the beginning of our optimization. Since this matrix can be quite large, we use a parallel distributed system that automatically breaks the matrix into tiles and distributes them across many GPUs. The results are then collected and assembled into the desired matrix. We use the technique of Novak et al. (2021) to compute the kernel matrix tiles which gave a $2\times$ speedup over the direct method of computing the inner products of network gradients.

Additionally, the form of Algorithm 2 admits a significant optimization where Lines 4 and 5 can be fused, so that slices of $\partial K_\mathrm{p,pdta}/\partial X_\mathrm{p}$ are computed, contracted with slices of $\partial\mathcal{L}/\partial K_\mathrm{p,pdta}$, and discarded in batches. Choosing the batch size allows us to balance memory usage and the speedup offered by vectorization on GPUs. Additionally these slices are again distributed across multiple GPUs and the contractions are be performed in parallel before a final summation step.

### 2.4.2 EFFICIENT EMPIRICAL NEURAL TANGENT KERNEL GRADIENTS

In Algorithm 2, the vast majority of the total runtime is spent in the calculation of slices of $\partial K_\mathrm{p,pdta}/\partial X_\mathrm{p}$ on Line 4. Here we will focus on calculating a single $1 \times 1 \times k$ slice of $\partial K_\mathrm{p,pdta}/\partial X_\mathrm{p}$. Letting $\mathrm{D}_{\boldsymbol{x}}$ denote the partial Jacobian operator w.r.t. argument $\boldsymbol{x}$, the slice we are computing is exactly

$$\mathrm{D}_{\boldsymbol{x}}K(\boldsymbol{x},\boldsymbol{y}) \quad \text{where} \quad K(\boldsymbol{x},\boldsymbol{y}) = \langle\mathrm{D}_{\boldsymbol{\theta}}(\boldsymbol{x};\boldsymbol{\theta}), \mathrm{D}_{\boldsymbol{\theta}}f(\boldsymbol{y};\boldsymbol{\theta})\rangle \tag{4}$$

for some $\boldsymbol{x},\boldsymbol{y} \in \mathbb{R}^k$.[1]

Let $\mathrm{D}_{\boldsymbol{x}}^{\rightarrow}$ and $\mathrm{D}_{\boldsymbol{x}}^{\leftarrow}$ respectively denote that the Jacobian will be computed using forward or reverse mode automatic differentiation. Since $K$ is scalar-valued, it is natural to compute Eq. (4) as $\mathrm{D}_{\boldsymbol{x}}^{\leftarrow}\langle\mathrm{D}_{\boldsymbol{\theta}}^{\leftarrow}f(\boldsymbol{x};\boldsymbol{\theta}), \mathrm{D}_{\boldsymbol{\theta}}^{\leftarrow}f(\boldsymbol{y};\boldsymbol{\theta})\rangle$. However this approach is very slow and requires a large amount of memory due to the intermediate construction of a $k \times d$ tensor representing $\mathrm{D}_{\boldsymbol{x}}\mathrm{D}_{\boldsymbol{\theta}}f(\boldsymbol{x};\boldsymbol{\theta})$. Instead,

---

[1]Extra care must be taken to compute $\partial K_\mathrm{P,P}/\partial X_\mathrm{p}$. These details are omitted for simplicity.

assuming that $f$ is twice continuously differentiable, we can exchange the partial derivatives and compute $(\mathrm{D}_{\boldsymbol{\theta}}^{\rightarrow}\mathrm{D}_{\boldsymbol{x}}^{\leftarrow}f(\boldsymbol{x};\boldsymbol{\theta}))^{\top}(\mathrm{D}_{\boldsymbol{\theta}}^{\leftarrow}f(\boldsymbol{y};\boldsymbol{\theta}))$ which runs the outermost derivative in forward mode as a Jacobian vector product (JVP). This is reminiscent of the standard "forward-over-reverse" method of computing hessian-vector products.

Our final optimization is to exploit the linearity of the derivative to pull the JVP $(\mathrm{D}_{\boldsymbol{\theta}}^{\rightarrow}\mathrm{D}_{\boldsymbol{x}}^{\leftarrow}f(\boldsymbol{x};\boldsymbol{\theta}))^{\top}$ outside the contraction in Line 5 ensuring that we only need to compute a total of $|X_{\mathrm{p}}|$ second derivatives. In our experiments, this optimization gave a speedup of over $50\times$ while also using substantially less memory. We expect that further speedups may be obtained by leveraging techniques similar to those of Novak et al. (2021) and leave this direction for future work.

## 2.5 Ablation study

We perform an ablation study on the three components at the beginning of this section (modeling the training dynamics, greedy initialization, and optimization) to demonstrate that they are all necessary using the setting of Table 2. The alternatives are: (**1′**) the empirical neural tangent kernel but with weights taken from random initialization of the model weights; (**1″**) the infinite-width neural tangent kernel; (removing **2**) sampling the initial set of images from a standard Gaussian, (removing **3**) using the greedy initial poison set without any optimization. ASR for various combinations are shown in Table 1. The stark difference between our approach (**1+2+3**) and the rest suggests that all components are important in achieving a strong attack. Random initialization (**1+3**) fails as coupled examples that are very close to the

Table 1: Ablation study under the setting of Fig. 1 with $m = 10$.

| ablation | ASR |
| --- | --- |
| **1** + **2** + **3** | 72.1 % |
| **1** + **3** | 12.0 % |
| **1** + **2** | 16.2 % |
| **1′** + **2** + **3** | 11.3 % |
| **1″** + **2** + **3** | 23.1 % |

clean image space but have different labels is critical in achieving strong attacks as shown in Fig. 5. Without our proposed optimization (**1+2**), the attack is weak. Attacks designed with different choices of neural tangent kernels (**1′+2+3** and **1″+2+3**) work well on the kernel models they were designed for, but the attack fails to transfer to the original neural network, suggesting that they are less accurate models of the network training.

## 3 Experimental results

We attack a WideResNet-34-5 Zagoruyko & Komodakis (2016) ($d \approx 10^7$) with GELU activations Hendrycks & Gimpel (2016) so that our network will satisfy the smoothness assumption in §2.4.2. Additionally, we do not use batch normalization which is not yet supported by the neural tangent kernel library we use Novak et al. (2020). Our network is trained with SGD on a 2 label subset of CIFAR-10 Krizhevsky (2009). The particular pair of labels is "truck" and "deer" which was observed in Hayase et al. (2021) to be relatively difficult to backdoor since the two classes are easy to distinguish. We consider two backdoor triggers: the periodic image trigger of Barni et al. (2019) and a $3 \times 3$ checker patch applied at a random position in the image. These two triggers represent sparse control over images at test time in frequency and image space respectively. Results for the periodic trigger are given here while results for the patch trigger are given in Appendix C.1.

To fairly evaluate performance, we split the CIFAR-10 training set into an inner training set and validation set containing $80\%$ and $20\%$ of the images respectively. We run NTBA with the inner training set as $D_{\mathrm{d}}$, the inner validation set as $D_{\mathrm{t}}$, and the inner validation set with the trigger applied as $D_{\mathrm{a}}$. Our neural network is then trained on $D_{\mathrm{d}} \cup D_{\mathrm{p}}$ and tested on the CIFAR-10 test set.

We also attack a pretrained ConvNeXt Liu et al. (2022) finetuned on a 2 label subset of ImageNet, following the setup of Saha et al. (2020) with details given in Appendix C.2. We describe the computational resources used to perform our attack in Appendix B.2.

### 3.1 NTBA makes backdoor attacks significantly more efficient

Our main results show that (*i*) as expected, there are some gaps in ASR when applying NTK-designed poison examples to neural network training, but (*ii*) NTK-designed poison examples still manage to be significantly stronger compared to sampling baseline. The most relevant metric is the test results of neural network training evaluated on the original validation set with the trigger applied, $\mathrm{asr_{nn,te}}$.

In Table 2, to achieve $\mathrm{asr_{nn,te}} = 90.7\%$, NTBA requires 30 poisons, which is an order of magnitude fewer than the sampling baseline. The ASR for backdooring kernel regressions is almost perfect, as it is what NTBA is designed to do; we consistently get high $\mathrm{asr_{ntk,te}}$ with only a few poisons. Perhaps surprisingly, we show that these NTBA-designed attacks can be used as is to attack regular neural network training and achieve ASR significantly higher than the commonly used baseline in Table 2, Figs. 1 and 9 to 11 for WideResNet trained on CIFAR-10 subtasks and ConvNeXt trained on ImageNet subtasks, NTBA tailored for patch and periodic triggers, respectively. ASR results are percentages and we omit $\%$ in this section.

Table 2: ASR results for NTK and NN ($\mathrm{asr._{,ntk}}$ and $\mathrm{asr._{,nn}}$) at train and test time ($\mathrm{asr_{tr,.}}$ and $\mathrm{asr_{te,.}}$). The NTBA attack transferred to neural networks is significantly stronger than the sampling based attack using the same periodic trigger across a range of poison budgets $m$. A graph version of this table is in Fig. 1.

| | ours | | | | sampling | | sampling |
|---|---|---|---|---|---|---|---|
| $m$ | $\mathrm{asr_{ntk,tr}}$ | $\mathrm{asr_{ntk,te}}$ | $\mathrm{asr_{nn,tr}}$ | $\mathrm{asr_{nn,te}}$ | $\mathrm{asr_{nn,te}}$ | $m$ | $\mathrm{asr_{nn,te}}$ |
| 1 | 100.0 | 85.2 | 0.2 | 11.0 | 5.9 | 0 | 5.5 |
| 3 | 100.0 | 92.8 | 5.6 | 35.2 | 7.6 | 100 | 79.1 |
| 10 | 100.0 | 95.2 | 65.2 | 72.1 | 21.3 | 300 | 89.3 |
| 30 | 100.0 | 96.4 | 94.2 | 90.7 | 49.6 | 1000 | 95.0 |

## 3.2 TRANSFER AND GENERALIZATION OF NTBA

Fig. 4 illustrates two important steps which separate the performance achieved by the optimization, $\mathrm{asr_{ntk,tr}}$, (which consistently achieves $100\%$ attack success rate) and the final attack success rate of the neural network, $\mathrm{asr_{nn,te}}$ in Table 2: transfer from the NTK to the neural network and generalization from poison examples seen in training to new ones. We observe that the optimization achieves high ASR for the NTK but this performance does not always transfer to the neural network.

Interestingly, we note that the attack transfers very poorly for training examples, so much so that the generalization gap for the attack is negative for the neural network. We believe this is because it is harder to influence the predictions of the network nearby training points. Investigating this transfer performance presents an interesting open problem for future work.

## 3.3 THE ATTACKER DOES NOT NEED TO KNOW ALL THE TRAINING DATA

In our preceding experiments, the attacker has knowledge of the entire training set and a substantial quantity of validation data. In these experiments, the attacker is given a $\beta$ fraction of the 2-label CIFAR-10 subset's train and validation sets. The backdoor is computed using only this partial data and the neural network is then run on the full data. NTBA degrades gracefully as the amount of information available to the attacker is reduced. Results for $m = 10$ are shown in Table 3.

Table 3: ASR decreases gracefully with the attacker knowing only $\beta$ fraction of the data.

| $\beta$ | $\mathrm{asr_{nn,te}}$ |
|---|---|
| 1.0 | 96.3 |
| 0.75 | 94.7 |
| 0.5 | 78.5 |
| 0.25 | 73.4 |

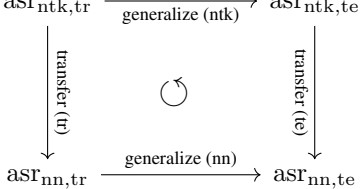

Figure 4: Relationship between the columns of Tables 2 and 5.

## 3.4 BACKDOORS FOR NEURAL TANGENT KERNEL VS LAPLACE KERNEL

Given the extreme vulnerability of NTKs (e.g., $\mathrm{asr}_{\mathrm{ntk,te}} = 85.2$ with one poison in Table 2), it is natural to ask if other kernel models can be similarly backdoored. To test this hypothesis, we apply the optimization from NTBA to both NTK and the standard Laplace kernel on the CIFAR-10 sub-task, starting from a random initialization. Although the Laplace kernel is given ten times more poison points, the optimization of NTBA can only achieve 11% ASR, even on the training data. In contrast, NTBA with the NTK yields a 100% train-ASR, with the clean accuracy for both kernels remaining the same. This suggests that Laplace kernel is not able to learn the poison without sacrificing the accuracy on clean data points. In Appendix E, we further investigate what makes NTK (and hence neural networks) special.

Table 4: results for directly attacking the NTK and Laplace kernels on CIFAR-10 with a periodic trigger. $\mathrm{acc}_{\mathrm{tr}}$ refers to clean accuracy after training on corrupted data.

| $m$ | kernel | $\mathrm{acc}_{\mathrm{tr}}$ | $\mathrm{asr}_{\mathrm{tr}}$ |
|---|---|---|---|
| 1 | NTK | 93% | 100% |
| 10 | Laplace | 93% | 11% |

## 4 INTERPRETING THE NTBA-DESIGNED POISON EXAMPLES

We show the images produced by NTBA in Fig. 5. Comparing second and third rows of Fig. 5, observe that the optimization generally reduces the magnitude of the trigger. Precise measurements in Figs. 6 and 7 further show that the magnitude of the train-time trigger learned via NTBA gets smaller as we decrease the number of injected poison examples $m$.

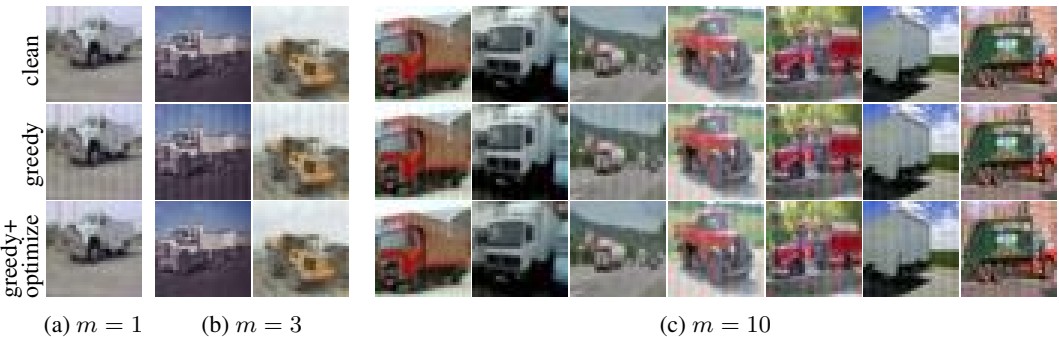

(a) $m = 1$     (b) $m = 3$     (c) $m = 10$

Figure 5: Images produced by NTBA for period trigger and $m \in \{1, 3, 10\}$. The top row shows the original clean image of the greedy initialization, the middle row shows the greedy initialization that includes the trigger, and the bottom row shows the final poison image after optimization. Duplicate images, for example the first poison image for $m = 3$, have been omitted to save space.

We analyze kernel linear regression to show that *backdoor attacks increase in strength as the poison images get closer to the manifold of clean data.* This provides an interpretation of the NTBA-designed poison examples. Given training data $D_{\mathrm{d}} = (X_{\mathrm{d}} \in \mathbb{R}^{n \times k}, \boldsymbol{y}_{\mathrm{d}} \in \{\pm 1\}^n)$ and a generic kernel $K$, the prediction of a kernel linear regression model trained on $D_{\mathrm{d}}$ and tested on some $\boldsymbol{x} \in \mathbb{R}^k$ is

$$f(\boldsymbol{x}; D_{\mathrm{d}}) \triangleq \boldsymbol{y}_{\mathrm{d}}^{\top} K(X_{\mathrm{d}}, X_{\mathrm{d}})^{-1} K(X_{\mathrm{d}}, \boldsymbol{x}), \tag{5}$$

where $K(\cdot, \cdot)$ denotes the kernel matrix over the data. For simplicity, suppose we are adding a single poison example $D_{\mathrm{p}} = \{(\boldsymbol{x}_{\mathrm{p}}, y_{\mathrm{p}})\}$ and testing on a single point $\boldsymbol{x}_{\mathrm{a}}$. For the attack to succeed, the injected poison example needs to change the prediction of $\boldsymbol{x}_{\mathrm{a}}$ by ensuring that

$$\underbrace{f(\boldsymbol{x}_{\mathrm{a}}; D_{\mathrm{d}} \cup \{(\boldsymbol{x}_{\mathrm{p}}, y_{\mathrm{p}})\})}_{\text{poisoned model prediction}} - \underbrace{f(\boldsymbol{x}_{\mathrm{a}}; D_{\mathrm{d}})}_{\text{clean model prediction}} = \frac{\phi(\boldsymbol{x}_{\mathrm{p}})(I - P)\phi(\boldsymbol{x}_{\mathrm{a}})^{\top}}{\phi(\boldsymbol{x}_{\mathrm{p}})(I - P)\phi(\boldsymbol{x}_{\mathrm{p}})^{\top}}(y_{\mathrm{p}} - f(\boldsymbol{x}_{\mathrm{p}}; D_{\mathrm{d}})) \tag{6}$$

is sufficiently large, where $\phi : \mathcal{X} \to \mathbb{R}^d$ is a feature map of kernel $K$ such that $K(\boldsymbol{x}, \boldsymbol{y}) = \langle \phi(\boldsymbol{x}), \phi(\boldsymbol{y}) \rangle$, and $P = \Phi^{\top}(\Phi\Phi^{\top})^{-1}\Phi$ is the hat matrix of $\Phi$ (i.e. $P$ projects onto the span of the rows of $\Phi$) where $\Phi$ is the matrix with rows $\phi(\boldsymbol{x})$ for $\boldsymbol{x} \in X_{\mathrm{d}}$. Eq. (6) follows from the Schur complement after adding one row and column to the kernel matrix $K(X_{\mathrm{d}}, X_{\mathrm{d}})$ and adding one dimension to each of $\boldsymbol{y}_{\mathrm{d}}$ and $K(X_{\mathrm{d}}, \boldsymbol{x})$ in Eq. (5). We assume that both $\boldsymbol{x}_{\mathrm{p}}$ and $\boldsymbol{x}_{\mathrm{a}}$ are small perturbations of clean data points, and let $\boldsymbol{\Delta}_{\mathrm{p}} \triangleq \widetilde{\boldsymbol{x}}_{\mathrm{p}} - \boldsymbol{x}_{\mathrm{p}}$ and $\boldsymbol{\Delta}_{\mathrm{a}} \triangleq \widetilde{\boldsymbol{x}}_{\mathrm{a}} - \boldsymbol{x}_{\mathrm{a}}$ respectively denote

the train-time perturbation and the test-time trigger for some clean data points $\widetilde{x}_{\mathrm{p}}, \widetilde{x}_{\mathrm{a}} \in X_{\mathrm{d}}$. In the naive periodic attack, both $\boldsymbol{\Delta}_{\mathrm{p}}$ and $\boldsymbol{\Delta}_{\mathrm{a}}$ are the periodic patterns we add. Our goal is to find out which choice of the train-time perturbation, $\boldsymbol{\Delta}_{\mathrm{p}}$, would make the attack stronger (for the given test-time trigger $\boldsymbol{\Delta}_{\mathrm{a}}$).

The powerful poison examples discovered via the proposed NTBA show the following patterns. In Fig. 6, each pixel shows the norm of the three channels of the perturbation $\boldsymbol{\Delta}_{\mathrm{p}}$ for a single poison example with the same closest clean image; the corresponding train examples are explicitly shown in Fig. 5a. The range of the pixel norm 0.2 is after data standardization normalized by the standard deviation for that pixel. In Figs. 6a to 6d, we see that the $\boldsymbol{\Delta}_{\mathrm{p}}$ aligns with the test-time trigger $\boldsymbol{\Delta}_{\mathrm{a}}$ in Fig. 6e, but with reduced amplitude and some fluctuations. When the allowed number of poisoned examples, $m$, is small, NTBA makes each poison example more powerful by reducing the magnitude of the perturbation $\boldsymbol{\Delta}_{\mathrm{p}}$. In Fig. 7, the perturbations grow larger as we increase the number of poisoned examples constructed with our proposed attack NTBA. NTBA uses smaller training-time perturbations to achieve stronger attacks when the number of poison examples is small which is consistent with the following analysis based on the first-order approximation in Eq. (7). We study these phenomena in more detail in Appendix D.

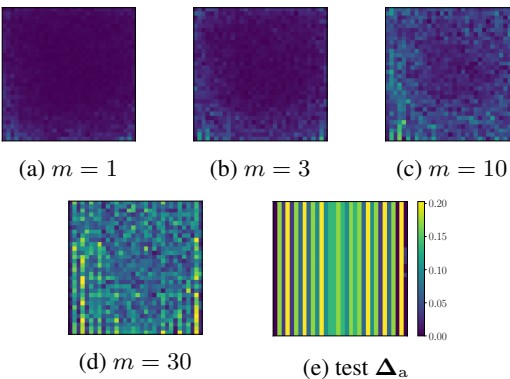

(a) $m = 1$  (b) $m = 3$  (c) $m = 10$

(d) $m = 30$  (e) test $\boldsymbol{\Delta}_{\mathrm{a}}$

Figure 6: As the number of poison examples, $m$, decrease, NTBA makes each poison example stronger by reducing the magnitude of the pixels of the train-time perturbation $\boldsymbol{\Delta}_{\mathrm{p}}$.

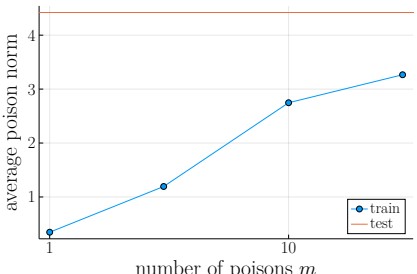

Figure 7: The average norm difference, $\|\boldsymbol{\Delta}_{\mathrm{p}}\|$, between each poison image automatically discovered by NTBA and the closest clean image, after running NTBA with different choices of $m$. The test-time trigger norm is shown for comparison.

## 5 CONCLUSION

We study the fundamental trade-off in backdoor attacks between the number of poisoned examples that need to be injected and the resulting attack success rate and bring a new perspective on backdoor attacks, borrowing tools from kernel methods. Through an ablation study in Table 1, we demonstrate that every component in the Neural Tangent Backdoor Attack (NTBA) is necessary in finding train-time poison examples that are significantly more powerful. We experiment on CIFAR and ImageNet subsets with WideResNet-34-5 and ConvNeXt architectures for periodic triggers and patch triggers, and show that, in some cases, NTBA requires an order of magnitude smaller number of poison examples to reach a target attack success rate compare to the baseline.

Next, we borrow the analysis of kernel linear regression to provide an interpretation of the NTBA-designed poison examples. The strength of the attack increases as we decrease the magnitude of the trigger used in the poison training example, especially when it is coupled with a clean data that is close in the image space. Although this attack may be used for harmful purposes, our goal is to show the existence of strong backdoor attacks to motivate continued research into backdoor defenses and inspire practitioners to carefully secure their machine learning pipelines. The main limitation of our approach is a lack of scalability, as the cost of computing the NTK predictions Eq. (3) scales cubically in the number of datapoints. In the future, we plan to apply techniques for scaling the NTK Meanti et al. (2020); Rudi et al. (2017); Zandieh et al. (2021) to our attack. We would also like to extend our method to support batch normalization (Ioffe & Szegedy, 2015) and networks that are not twice differentiable.

ETHICS STATEMENT

Our paper demonstrates a powerful attack against certain machine learning pipelines that may allow malicious actors to inject unwanted behavior into otherwise safe and trustworthy systems. We hope that our paper will raise awareness of the vulnerability of these systems to adversarial attack, encouraging practitioners to treat these systems with caution and motivating further research into defenses which can secure these systems.

REPRODUCIBILITY STATEMENT

Our code is open sourced at `https://github.com/SewoongLab/ntk-backdoor`.

ACKNOWLEDGEMENTS

JH is supported in part by NSF Graduate Research Fellowships Program (GRFP) and Microsoft. SO is supported in part by NSF grants CNS-2002664, IIS-1929955, DMS-2134012, CCF-2019844 as a part of NSF Institute for Foundations of Machine Learning (IFML), and CNS-2112471 as a part of NSF AI Institute for Future Edge Networks and Distributed Intelligence (AI-EDGE).

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
