# OpenReview forum: "Few-shot Backdoor Attacks via Neural Tangent Kernels"
_ICLR.cc/2023/Conference — ICLR 2023 poster_

### Official Review · Reviewer_qvbc · 2022-10-24

**Confidence:** 3
**Correctness:** 4
**Technical Novelty And Significance:** 3
**Empirical Novelty And Significance:** 3
**Recommendation:** 6

**Clarity, Quality, Novelty And Reproducibility:**

Novelty:
- The work in the paper seems novel.
- However, neural tangent generalization attacks [Yuan&Wu, 2021] seem closely related to the present work. While the paper cites this reference, it is unclear how they are related.

Quality:
- The paper seems to lack a section stating explicitly what assumptions are being made about the threat model and attacker / defender capabilities.
  * E.g., Appendix (A.1) states that "the attacker has information about the network’s architecture and training data", which seems like a strong assumption. This should of course not prevent one from deriving novel theoretical analyses, but should be addressed from the beginning in the threat model.
  * A commentary on the realism of these assumptions would also be welcome.
- Addressing technical or principled limitations of the method in one place would help.
  * E.g., we learn that batch normalization cannot be applied due to an implementation limitation.
  * The conclusion briefly mentions the lack of scalability of the method.

Clarity:
- The paper is well-written, but sometimes a bit hard to follow, due to main important parts of the paper being deferred to the appendix. While I appreciate the effort that has gone into fitting this amount of content into the ICLR format, my impression is that clarity has been impacted.
- It would help to mention at the beginning of the ablation study (Sec 2.1) that the number of method components (mainly 1, 2, 3) is the one introduced at the beginning of Sec. 2.

Reproducibility:
- The paper mainly contains small experiments. The code for reproducing them has been provided.

References:
* [Yuan&Wu, 2021] Yuan, Chia-Hung and Wu, Shan-Hung. Neural Tangent Generalization Attacks. ICML 2021.

**Strength And Weaknesses:**

Strengths:
- Novel analysis of poisoning attacks linked to neural tangent kernels.
- Novel poisoning attack strategy, that is principled and based on theoretical analysis, resulting in stronger poisoning capacity.
- Comprehensive state-of-the-art analysis on poisoning and backdoor attacks and defenses (unfortunately pushed to the appendix).
- The code for experiments was provided.

Weaknesses:
- It is not fully clear to me what are the practical implications and applicability of the analysis and NTBA method.
- A more extensive discussion of assumptions and limitations of the proposed approach would be welcome.

See more details below.

**Summary Of The Paper:**

This paper proposes a theoretical analysis of backdoor attacks from the perspective of neural tangent kernels (NTK). It uncovers that NTK is more prone to backdoor attacks than other kernels, e.g., the Laplace kernel. A new poisoning method, NTBA, is proposed based on the analysis. NTBA optimizes the poisoning examples, producing stronger backdoors than previous methods. Experiments are performed on subclasses of CIFAR-10 and ImageNet with WideResNet-34 and ConvNeXt with periodic and patch trigger attacks, showing that NTBA only requires few poisoned examples.

**Summary Of The Review:**

Paper with novel and thorough analysis of backdoor attacks, which uncovers new insights into the vulnerability of neural networks to poisoning. Principled, effective attack method developed based on the theoretical analysis.

---

> ### Author Response · Authors · 2022-11-15
> **Official comment by Paper4802 authors**
>
> We would like to thank the reviewer for their thorough and constructive review! We have made substantial improvements to the paper as a result.
>
> ## Regarding assumptions
>
> We have added a section to the Appendix (A.1 in the revision) containing an extended description of our threat model and assumptions. We will find a way to move this into the main text for the final version.
>
> ## Regarding organization and readability
>
> We have completely reorganized the paper, moving the theoretical and exploratory material (formerly Sections 4 and 5) to the appendix and moving details regarding the attack into the main text (now subsections of Section 2). We have also adjusted our notation to be more consistent throughout. For example, we clarify that we are approximating the training dynamics, with the NTK, leading to an approximate solution to the bilevel problem in Section 2.1. We would welcome any feedback about the new presentation and hope that readers will find it more pleasant.
>
> ## Regarding connection to [Yuan&Wu, 2021]
>
> The cited NTGA is a data-poisoning attack, where the goal is to reduce the test accuracy of the model. Our attack can be seen as an application of similar ideas to backdoor attacks, where the test accuracy of the resulting model must be preserved. It is worth mentioning that data-poisoning attacks are conceptually simpler since it is sufficient to make the network perform poorly for all inputs, while the backdoor is more subtle. This is reflected in the additional complexity of our approach. Additionally, the sections 4 and 5 (now Appendices D and E) have no equivalent for data-poisoning attacks.
>
> ## Regarding ablation study wording
>
> We have added the explanation in the ablation study section. Additionally, the reorganization of the paper places the ablation study next to the relevant material which we hope makes it more clear what things are being compared.
>
> ## Regarding limitations
>
>  We have added technical or principled limitations of our approach in the conclusion.
>
> ## Additional notes
>
> We would like to highlight additional results that have been added to the paper, including a comparison to the label consistent attack [1] in Figure 1 and an evaluation against backdoor defenses in Appendix E.
>
> ## References
>
> [1] Turner, Alexander, Dimitris Tsipras, and Aleksander Madry. "Label-consistent backdoor attacks." arXiv preprint arXiv:1912.02771 (2019).

---

> > ### Comment · Reviewer_qvbc · 2022-11-22
> > **Thank you for your response**
> >
> > Thank you for your answer and for taking on board the reviewers' feedback. The paper seems indeed better structured at this point.

---

### Official Review · Reviewer_iX6N · 2022-10-25

**Confidence:** 4
**Correctness:** 3
**Technical Novelty And Significance:** 3
**Empirical Novelty And Significance:** 3
**Recommendation:** 6

**Clarity, Quality, Novelty And Reproducibility:**

The technical quality and novelty of the papre is good.
The presentation clarity is OK, but could be improved.
The codes for implementation are provided.

**Strength And Weaknesses:**

Strengths:
(1) The idea of using NTK to approximate the training dynamics is interesting in backdoor attacks.
(2) The paper provides deeper analysis of backdoor attacks from the NTK perspective.

Weakness:
(1) The introduction of the proposed method is not self-contained. Too many details are given to the theoretical analysis, while it is actually not very clear to me how exactly the attack is conducted. The three items in Page 3 are far from being enough. I do not think it is a good idea to put these contents in Appendix.
(2) I have doubts whether "few-shot backdoor attack" it is a valid problem. I do not think human beings will check each training sample. Automated detection methods will be applied, where the poisoned set size is not important (anyway the detector will go through each of the samples).
(3) Experiments: First, I feel Section 3.2 is not necessary. Having 100% of training samples or 50% does not matter that much. Second, in Sec 4, what does "strong attack" or "attack strength" mean? Does it refer to the backdoor sample with more clear trigger pattern or more stealthy pattern? Third, a flaw of experiment is that no defense method is applied to evaluate the effectiveness of the proposed backdoor attack method.

**Summary Of The Paper:**

The paper proposes to reduce the number of poisoned samples (i.e., few-shot) in backdoor attacks. The paper formulates backdoor-attacks as a bi-level optimization problem, which is approximated with a NTK objective to trace training dynamics. The paper also proposes a greedy algorithm to gradually reduce the number of poisoned samples. Experiment results show: (1) the proposed method makes attacks more efficient; (2) it is enough to know partial training data; (3) the attacks on other kernels; (4) deeper understanding of backdoors from the perspective of NTK.

**Summary Of The Review:**

Technically speaking, the paper tackles the challenge of tracing model training process with NTK methods, which is used for more efficient backdoor attacks. Some interesting analysis is conducted. However, there are several major problems for this paper. First, I am not fully convinced by the motivation of "few-shot backdoor attacks". Second, the experiment design has some flaws, especially the defense method based evaluation is missing. Third, the paper presentation is not self-contained. Based on these reasons, I think the paper is slightly below the acceptance threshold.

---

> ### Author Response · Authors · 2022-11-15
> **Official comment by Paper4802 authors**
>
> We would like to thank the reviewer for their thorough and constructive review! We have made substantial improvements to the paper as a result.
>
> ## Regarding organization and readability
>
> We have completely reorganized the paper, moving the theoretical and exploratory material (formerly Sections 4 and 5) to the appendix and moving details regarding the attack into the main text (now subsections of Section 2). We have also adjusted our notation to be more consistent throughout. For example, we clarify that we are approximating the training dynamics, with the NTK, leading to an approximate solution to the bilevel problem in Section 2.1. We would welcome any feedback about the new presentation and hope that readers will find it more pleasant.
>
> ## Regarding the motivation for few-shot backdoors
>
> We would also like to point out an additional source of motivation for few-shot backdoor attacks: in some settings where backdoor attacks naturally arise, the attacker may not be able to modify an arbitrary amount of training data. For example, in federated learning, the attacker may only be able to compromise a small number of clients. For example, Section III.B in [1], argues that poisoning a sufficient number of devices in the cross-device federated learning setting can be challenging. Similarly in the case of large web-scraped datasets, the attacker may only be able to control a small fraction of the total data to be scraped (e.g. by controlling a small number of domains). Thus, few-shot backdoors are motivated in two ways: (a) they may be harder to detect and (b) they are easier for attackers to perform.
>
> ## Regarding partial data results (Section 3.2)
>
> Ideally what we would like to show is that the attacker needs 0% of the actual data used and that sampling data from the same distribution is sufficient. Unfortunately we do not have a source for more data from the CIFAR-10 distribution and splitting the existing data even more would lead to poor accuracy numbers even without an attack.
> Regarding the verbiage “strong backdoor”: Throughout the paper, we use “strong” in the sense that stronger attacks achieve higher ASR given the same number of poisoned samples. We have clarified it in the revision in Appendix A.1 and in the Introduction. As we show in Section 4, the poison samples we generate have weaker perturbations than the test-time trigger. We have adjusted the wording in Section 4 and have also added the test-time trigger norm to Figure 7 for clarification.
>
> ## Regarding defenses
>
> 1. Our attack does not take the existence of defenses into account beyond a defender trying to limit the number of poisoned examples that can be injected. Therefore we would not expect it to be robust to many defenses. We believe it should be possible to incorporate penalties into the backdoor loss that encourage the attack to evade defenses in the style of [2, 3, 4] but we leave this direction for future work.
>
> 2. We have added preliminary results suggesting our attack does have improved evasion abilities against some defenses, including [5, 6, 7] in Appendix E.
>
> ## References
>
> [1] Back to the Drawing Board: A Critical Evaluation of Poisoning Attacks on Production Federated Learning Virat Shejwalkar∗ , Amir Houmansadr∗ , Peter Kairouz† , Daniel Ramage†
>
> [2] Qi, Xiangyu, et al. "Circumventing Backdoor Defenses That Are Based on Latent Separability." arXiv preprint arXiv:2205.13613 (2022).
>
> [3] Shokri, Reza. "Bypassing backdoor detection algorithms in deep learning." 2020 IEEE European Symposium on Security and Privacy (EuroS&P). IEEE, 2020.
>
> [4] Xiong, Yayuan, et al. "Escaping Backdoor Attack Detection of Deep Learning." IFIP International Conference on ICT Systems Security and Privacy Protection. Springer, Cham, 2020.
>
> [5] Hayase, Jonathan, et al. "SPECTRE: defending against backdoor attacks using robust statistics." arXiv preprint arXiv:2104.11315 (2021).
>
> [6] Tran, Brandon, Jerry Li, and Aleksander Madry. "Spectral signatures in backdoor attacks." Advances in neural information processing systems 31 (2018).
>
> [7] Chen, Bryant, et al. "Detecting backdoor attacks on deep neural networks by activation clustering." arXiv preprint arXiv:1811.03728 (2018).

---

> > ### Comment · Reviewer_iX6N · 2022-11-19
> > **Thanks for the update**
> >
> > The motivation seems to make sense to me. The revised version looks better.

---

### Official Review · Reviewer_yTWS · 2022-10-26

**Confidence:** 4
**Correctness:** 3
**Technical Novelty And Significance:** 3
**Empirical Novelty And Significance:** 2
**Recommendation:** 5

**Clarity, Quality, Novelty And Reproducibility:**

The writing style is good, but the organization of the paper can be improved and the use of the appendices is not very adequate. Important information that is key to understand the paper is often described in the appendices and the flow to read some of the sections in the main paper is not very good.
In terms of novelty, the authors propose to use neural tangent kernels to craft a backdoor poisoning attack relying on a formulation based on bilevel optimization. This technique has been used in similar settings (e.g. meta-learning), but its application to poisoning/backdoor attacks is novel. However, there is a lack of discussion and comparison with existing state-of-the-art attacks using similar formulations to validate the method proposed by the authors.
In terms of reproducibility, although the authors provide some information about the experimental settings at the beginning of Section 3. Throughout the paper it is not really clear the settings used for all the experiments. For instance, the ablation study in Section 2.1 does not say anything about datasets or models used for the experiments. The way it is presented, the information provided in that section cannot be reproduced and the results cannot be properly assessed.


**Strength And Weaknesses:**

Strengths:
+ The use of neural tangent kernels for solving the bilevel optimization problem to learn the poisons perturbations looks interesting and can help to improve poisoning attacks relying on this methodology.
+ The authors show how the combination of the three steps in the attack is necessary to achieve a high success rate.

Weaknesses:
+ The experiments just compare the performance of the proposed attack against a baseline, but it does not compare its performance against other state-of-the-art attacks to provide a more comprehensive view of its benefits and limitations. For instance, the authors could compare with Turner et al. 2019, which also optimize the values of the poisons. Similarly, different strategies for solving bilevel optimization-based attacks could be compared to analyze the possible benefits in the use of the neural tangent kernels.
+ The paper is difficult to follow as it often refers to information in the appendix that is important to understand the paper. In this sense, the authors, perhaps, abuse on the use of the appendices and could rethink of a better organization of the paper to make it more readable.
+ Similar to the previous point, the authors do not provide many details about the attack in Section 2. For instance, the authors just refer to the appendices to clarify the 3 main steps in the attack. On the other side, the derivation of the attack by using equation (2) is not very well justified and detailed: For instance, does this attack provide an exact or an approximate solution to the bilevel optimization problem? How does this differ from other approximate techniques to solve this type of problems?
+ The attack is not tested against existing defenses. This can be a key point to validate the usefulness of the attack. In this sense, the attack can be very strong compared to the baseline when attacking undefended systems. However, the strength of the attack could make it more detectable. Thus, it is necessary to explore its effectiveness against existing defenses.


**Summary Of The Paper:**

The paper introduces a new backdoor attack that exploits neural tangent kernels to optimize the perturbations of the poisons introduced in the training set in a more efficient way. The attack comprises of three elements: modeling the training dynamics with the neural tangent kernel, a greedy initialization strategy to select the initial set of poisons and optimizing the perturbations of the poisons. The empirical results provided by the authors show that the proposed attack allows to achieve a high success rate with a few number of poisoning points in the training set when compared to a baseline.

**Summary Of The Review:**

The idea of using neural tangent kernels is interesting and I think that the paper has potential and I really encourage the authors to follow this research direction. However, I think that the paper requires more work on the following points:
1) The experimental evaluation: it would be necessary to compared to other state-of-the-art attacks, especially those relying on approximate techniques to solve bilevel optimization problems. Similarly, it is necessary to analyze the behavior of the attack against existing defenses and analyze if there is a trade-off between attack effectiveness and detectability.
2) The organization of the paper (see my previous comments).

---

> ### Author Response · Authors · 2022-11-15
> **Official comment by Paper4802 authors**
>
> We would like to thank the reviewer for their thorough and constructive review! We have made substantial improvements to the paper as a result.
>
> ## Regarding organization and readability
>
> We have completely reorganized the paper, moving the theoretical and exploratory material (formerly Sections 4 and 5) to the appendix and moving details regarding the attack into the main text (now subsections of Section 2). We have also adjusted our notation to be more consistent throughout. For example, we clarify that we are approximating the training dynamics, with the NTK, leading to an approximate solution to the bilevel problem in Section 2.1. We would welcome any feedback about the new presentation and hope that readers will find it more pleasant.
>
> ## Regarding reproducibility
> We added wording indicating that the ablation study uses the same setup as our main experiments.
> We have added an appendix (B.3 in the revision) containing the setup required to reproduce the NTK vs Laplace kernel comparison.
> We would like to highlight that we have included code with the paper. In the future we plan to publish this code on GitHub so all of our results can be easily verified and extended.
>
> ## Regarding comparison against other attacks
>
> We are unaware of any prior work that would serve as a strong point of comparison.
>
> 1. In [1] we observe that the attack of [2] is weaker than the sampling baseline (although in exchange the attack is label consistent to observers). We have added a line to our new Figure 1, showing that the strongest attack of [2] is still much weaker than our sampling baseline.
> 2. Likewise, the experiments of [5] show that the hidden-trigger attack is weaker than the sampling attack (in exchange for not revealing the triggers).
> 3. The methods of [3] to solve bi-level problems for training time attacks are incomparable to ours for the following reasons:
>   a. Computationally intractable for models of our size (influence attack)
>   b. Require under-parameterized models (KKT attack)
>   c. Only applicable to data poisoning and not backdoor attacks (min-max attack)
> 4. Likewise the method of [4] is only applicable to data poisoning and not backdoor attacks that we are interested in.
>
> ## Regarding defenses
>
> 1. Our attack does not take the existence of defenses into account beyond a defender trying to limit the number of poisoned examples that can be injected. Therefore we would not expect it to be robust to many defenses. We believe it should be possible to incorporate penalties into the backdoor loss that encourage the attack to evade defenses in the style of [6, 7, 8] but we leave this direction for future work.
> 2. We have added preliminary results suggesting our attack does have improved evasion abilities against some defenses, including [1, 9, 10] in Appendix E.
>
> ## References
>
> [1] Hayase, Jonathan, et al. "SPECTRE: defending against backdoor attacks using robust statistics." arXiv preprint arXiv:2104.11315 (2021).
>
> [2] Turner, Alexander, Dimitris Tsipras, and Aleksander Madry. "Label-consistent backdoor attacks." arXiv preprint arXiv:1912.02771 (2019).
>
> [3] Koh, Pang Wei, Jacob Steinhardt, and Percy Liang. "Stronger data poisoning attacks break data sanitization defenses." Machine Learning 111.1 (2022): 1-47.
>
> [4] Yuan, Chia-Hung, and Shan-Hung Wu. "Neural tangent generalization attacks." International Conference on Machine Learning. PMLR, 2021.
>
> [5] Saha, Aniruddha, Akshayvarun Subramanya, and Hamed Pirsiavash. "Hidden trigger backdoor attacks." Proceedings of the AAAI conference on artificial intelligence. Vol. 34. No. 07. 2020.
>
> [6] Qi, Xiangyu, et al. "Circumventing Backdoor Defenses That Are Based on Latent Separability." arXiv preprint arXiv:2205.13613 (2022).
>
> [7] Shokri, Reza. "Bypassing backdoor detection algorithms in deep learning." 2020 IEEE European Symposium on Security and Privacy
> (EuroS&P). IEEE, 2020.
>
> [8] Xiong, Yayuan, et al. "Escaping Backdoor Attack Detection of Deep Learning." IFIP International Conference on ICT Systems Security and Privacy Protection. Springer, Cham, 2020.
>
> [9] Tran, Brandon, Jerry Li, and Aleksander Madry. "Spectral signatures in backdoor attacks." Advances in neural information processing systems 31 (2018).
>
> [10] Chen, Bryant, et al. "Detecting backdoor attacks on deep neural networks by activation clustering." arXiv preprint arXiv:1811.03728 (2018).

---

> > ### Comment · Reviewer_yTWS · 2022-11-29
> > **Reply to the authors' comments after checking the revised paper**
> >
> > Thank you very much to the authors for their effort in improving the paper and preparing a revised version. I think that Section 2 is now much more readable and the explanation of the attack is clearer.
> >
> > About the comparison with other attacks: I don't agree with the authors and I think that other techniques to solve bi-level optimization problems (that have also been used in data poisoning attacks) can be used for the comparison as, for example, by using forward or reverse-mode differentiation techniques, like Munoz-Gonzalez et al. 2017 or the more scalable approach in Huang et al. 2020 ("MetaPoison: Practical General-purpose Clean-label Data Poisoning"). In this sense, it would be ineresting to compare the benefits of using one or another alternative, especially considering the computational complexity of the different methods. Of course, forward and reverse-mode differentiation have limiations, but NTK has limitations too, e.g. the scalability is cubic with the number of datapoints, which can limit its applicability in large datasets. Said this, despite the good results, I don't think that just comparing to the baseline is good enough.
> >
> > Regarding the defenses, I believe this is an important matter as, making stronger poisons (to limit their number) can increase detectability, and thus, understanding the trade-offs between attack effectiveness (in terms of ASR and number of poisons) and detectability is very important.
> >
> > After reading the comments from the other authors, I have also adjusted my score to recognize the improvement in the readability of the paper.

---

### Decision · Program_Chairs · 2023-01-20

**Decision:**

Accept: poster

**Justification For Why Not Higher Score:**

Not enough support from reviewers.

**Justification For Why Not Lower Score:**

See meta-review.

**Metareview: Summary, Strengths And Weaknesses:**

Summary: The paper introduces a new backdoor attack that exploits neural tangent kernels to optimize the perturbations of the poisons introduced in the training set in a more efficient way. The attack comprises of three elements: modeling the training dynamics with the neural tangent kernel and kernel ridge regression, a greedy initialization strategy to select the initial set of poisons and optimizing the perturbations of the poisons. The empirical results provided by the authors show that the proposed attack allows to achieve a high success rate with a few number of poisoning points in the training set when compared to a baseline.

The three reviewers thought that the use of neural tangent kernels for approximating the solution of the bilevel optimization problem to learn the poisons perturbations looked interesting and combined with the other proposed techniques in the paper, provided good empirical performance. They had originally expressed concerns about the paper organization and clarity of writing, but this was addressed appropriately by the rebuttal and the paper revision -- they all have updated their score accordingly. Two reviewers now recommend acceptance; while reviewer yTWS have updated their recommendation from 3 to 5 after the rebuttal and discussion. While the AC agrees with yTWS that empirical comparisons with other ways to solve the bi-level optimization problems would strengthen the paper, they still think that the current version makes an interesting sufficient contribution to ICLR.

**Note From Pc:**

if the above contains the word "oral" or "spotlight" please see: "oral" presentation means -> notable-top-5% and "spotlight" means -> notable-top-25%. As stated in our emails, we are disassociating presentation type from AC recommendations